

# Rehabilitation time has greater influences on soil mechanical composition and erodibility than does rehabilitation land type in the hilly-gully region of the Loess Plateau, China

Leilei Qiao[1,*], Wenjing Chen[1,3,*], Yang Wu[3], Hongfei Liu[3], Jiaoyang Zhang[4,5], Guobin Liu[1,4] and Sha Xue[2,3,4]

[1] State Key Laboratory of Soil Erosion and Dryland Farming on the Loess Plateau, Institute of Soil and Water Conservation, Northwest A&F University, Yangling, China
[2] Shaanxi Key Laboratory of Land Consolidation, Chang'an University, Xi'an, China
[3] Collage of Forestry, Northwest A&F University, Yangling, China
[4] Institute of Soil and Water Conservation, Chinese Academy of Sciences and Ministry of Water Resources, Yangling, China
[5] University of Chinese Academy of Sciences, Beijing, China
[*] These authors contributed equally to this work.

Corresponding author
Sha Xue, xuesha100@163.com

## ABSTRACT

**Background**. The major landscape in the hilly-gully region of the Loess Plateau is greatly affected by vegetation rehabilitation on abandoned cropland. Although many studies have shown that the rehabilitation have greatly improved soil conditions and protected them from erosion, these effectiveness were not always in consensus possibly due to the land type of vegetation or to the rehabilitation time. To close this gap, we conducted a long term experiment as follows.

**Methods**. In this study, we analysed four land types of vegetation rehabilitation (shrub land, woodland, naturally revegetated grassland, and orchard land) with different rehabilitation times and investigated the mechanical composition and erodibility of the soil. Areas of slope croplandand natural forest were selected as controls.

**Results**. The results showed that soil depth, rehabilitation time and rehabilitation land type had strong impacts on soil mechanical composition, micro-aggregation and erodibility. Following rehabilitation, naturally revegetated grassland and shrub land had lower fractal dimensions of particle size distribution (fractal dimensions of PSD), fractal dimensions of micro-aggregation, and erodibility (K factor) than did cropland. Compared to the positive effects of rehabilitation mainly happened in the topsoil layer at other rehabilitation land type, that of woodland happened in the deeper soil layer. Besides, the indispensable rehabilitation time for the significant improvement of soil condition was shorter at naturally revegetated grassland than that at shrub land and woodland.

**Discussion**. Although rehabilitation time was more influential than was rehabilitation land type or soil depth, the differences among the rehabilitation land types showed that naturally revegetated grassland with native plants is the most time-saving rehabilitation vegetation for the Loess Plateau in the conversion from slope cropland. The success of rehabilitation in this forestry practice was mainly contributed by the suited species

of rehabilitation land type to the local climate and soil. Based on the differences of rehabilitation effectiveness resulting from land type, we should be cautious to choose land types for the rehabilitation of soil conditions in the Loess Plateau.

## INTRODUCTION

Soil erosion initiated by either natural or human factors is a serious environmental problem in many parts of the world. It not only caused the degradation of soil quality but also the destruction of ecosystem function and safety (*Chen & Peng, 2000*; *Lian et al., 2013*; *Borrelli et al., 2017*). Severe soil erosion is a serious challenge in the Loess Plateau of China (*Sun, Liu & Xue, 2016b*). To alleviate soil erosion and restore the local ecological environment, the Chinese government implemented the "Grain for Green" programme in 1999 to convert degraded cropland to forest and grassland (*Uchida, Xu & Rozelle, 2005*; *Chen et al., 2007a*; *Uchida, Rozelle & Xu, 2009*; *Zhang et al., 2011a*; *Song et al., 2015*). This programme has greatly decreased soil loss (*Chen et al., 2007b*).

Several studies had examined the effects of plant species changes, land preparation, rainfall intensity, anthropogenic disturbance, afforestation, and land abandonment on the mechanical composition and erodibility of the soil (*Koulouri & Giourga, 2007*; *Keesstra et al., 2009*; *Xia et al., 2009*; *Zhang et al., 2011b*; *Moora et al., 2014*; *Yu et al., 2017*). Soil mechanical composition and micro-aggregate stability were shaped by complicated geophysical and environmental processes and responded to land-use changes, thereby affecting soil hydrological and mechanical functioning and soil erosion (*Wang, Liu & Liu, 2005*; *Alagöz & Yilmaz, 2009*; *Xiao et al., 2014*; *Wang et al., 2016*). Many studies have reported positive impacts of vegetation rehabilitation of sloped croplandon soil conditions and soil resistance to erosion (*Xu, Li & Li, 2013*; *Ziadat & Taimeh, 2013*; *Xiao et al., 2014*; *Fu et al., 2015*; *Sun, Liu & Xue, 2016b*). Different plant species, with differences in morphology, architecture and other biological characteristics, show variation in their effectiveness for vegetation rehabilitation (*Bochet & García-Fayos, 2004*; *Ghestem et al., 2014*; *Fu et al., 2015*). However, local precipitation, parent material, disturbance and their interaction and sampling time can influence vegetation rehabilitation and make interpretation of results challenging. Thus, long-term research on the dynamics of soil erosion is necessary to understand the effects of vegetation rehabilitation on soil physical condition while accounting for confounding factors. However, several studies have focused on the effects of different rehabilitation patterns or the dynamic changes following rehabilitation in a certain land type but have not clearly identified the impacts of the various rehabilitation land type on the soil mechanical composition and erodibility during a long-time scale. Soil erosion, solution transformation and soil-moisture are influenced by soil particle size distribution (PSD) (*Mazaheri & Mahmoodabadi, 2012*; *Yu et al., 2015*). Land use could influence soil structure and physical and biochemical activity through PSD

affected by water erosion (*Basic et al., 2004*; *Su et al., 2004*). Therefore, its variation remains to be characterized for understanding and evaluating soil structure and dynamics and the effects of land use on soil structure. Fractal theory, an effective and reliable tool, can be used to characterize it (*Chen & Zhou, 2013*).

For the past fifty years, people pressurized by an increasing population into converting native grasslands into farmlands in the most parts of Loess Plateau of China, which caused the loss of most of the topsoil in many locations (*Wei et al., 2006*; *Zhou, Shangguan & Zhao, 2006*). The "Grain for Green" Programs (GGP) launched by Chinese government aimed at reducing soil erosion through replacing degraded cropland with forest and grassland. Since then, a sloped cropland was abandoned and restored naturally and artificially. As we know, herbs, rather than trees or shrubs, were dominant on the Loess Plateau due to its special geological characteristics in a long historical period. A proper choice of rehabilitation land type for the success of afforestation is the key thing (*Lü, Liu & Guo, 2003*; *Jiang et al., 2013*). Thus, the difference of this rehabilitation effectiveness between native vegetation (naturally revegetated native grass) and common forestry afforestation (artificial ecological forest, artificial economic forest, artificial shrub) deserves our attention. In this study, we collected comprehensive and long-term data on historic vegetation (e.g., forest, shrubland and grassland) with different rehabilitation times (1) to elucidate the effects of rehabilitation land type, time and soil depth on soil mechanical condition and erodibility; (2) to clearly identify the key influencing factors.

## MATERIAL AND METHODS

### Experimental area

This study was conducted in Ansai County, Shannxi Province, China (36°31′–37°20′N, 108°52′–109°26′E; 1,012–1,731 m a.s.l.), which lies in the middle part of the Loess Plateau. This region has a typical semiarid continental climate with a mean annual temperature of 8.8 °C, meaning that monthly temperature ranges from 22.5 °C in July to 7 °C in January and an annual precipitation of 549.1 mm, which mainly occurs between July and September (*Sun, Liu & Xue, 2016a*). The landform is characterized by a deeply incised hilly-gully Loess landscape. The soil in this area is mainly Huangmian soil, a Calcic Cambisol classified in the WRB reference system (*FAO/UNESCO/ISRIC, 1988*), originating from wind-blown deposits and characterised by yellow color, absence of bedding, silty texture, looseness, macroporosity, and wetness-induced collapsibility (*Xiao et al., 2014*). This type of soil is characterized by weak cohesion (*Sun, Liu & Xue, 2016a*), which makes it highly susceptible to severe soil erosion.

So far, native vegetation on the Loess Plateau remains controversial, because the vegetation cover of the Loess Plateau has been changing greatly during the historical period. The research about paleo-pedology, phytolith, organic carbon stable isotope and pollen records showed that herbs, rather than trees or shrubs, were dominant on the Loess Plateau in both the cold-dry period and the warm-humid period, owing to specific lithological property with thick loess which can not support an extensive forest development, even during the climatic optimum in this area (*Lü, Liu & Guo, 2003*; *Jiang et*

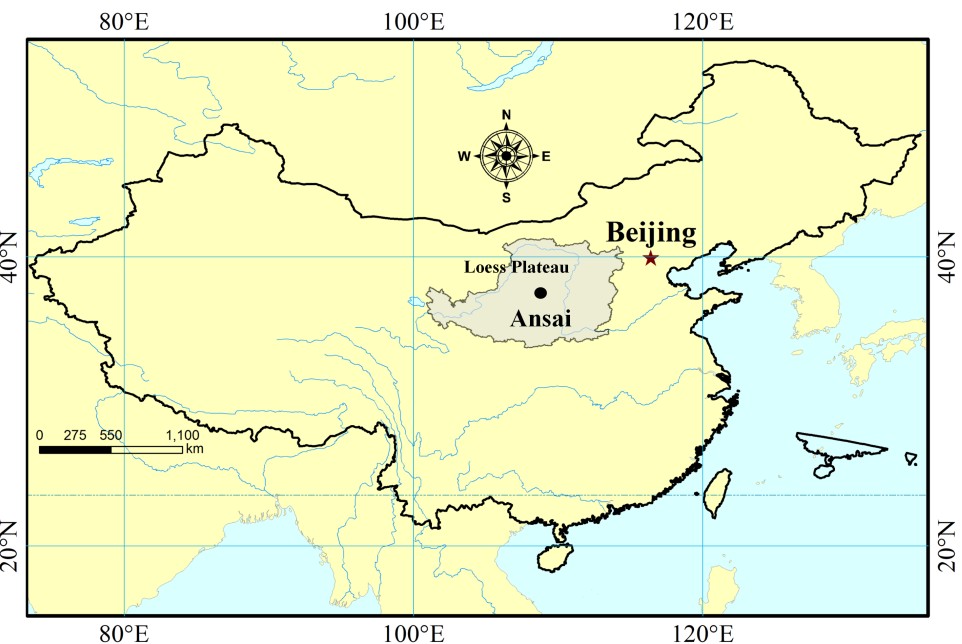

**Figure 1** Location of the Loess Plateau China.

*al., 2013*). Overgrazing, deforestation and other land-use patterns led to severe damage to the ecological environment and severe soil erosion by the middle of the last century. Since the late 1950s, the land use type in this region changed remarkably by the GGP, sloping cropland has been replanted with woodland (*Robinia pseudoacacia*), shrubland (*Caragana korshinskii, Hippophae rhamnoides*), artificial grassland (*Medicago sativa*) and naturally revegetated grassland to control soil erosion (*Sun, Liu & Xue, 2016b*). Much of barren lands and degraded croplands with slopes over 15° were rebuilt, accounting for about 14% of hilly Loess Plateau (*Xu, Wang & Zhao, 2018*). Artificially and naturally rebuilt grassland, shrub land, and planted woodland consist of the main land use types in the region.

In this region, before restoration the soil is weakly cohesive and thus prone to erosion, with erosion modulus of 10,000–12,000 $mg\,km^{-2}\,yr^{-1}$ (*Liu, 1999*). Recently, some soil and water conservation measure such as revegetation have effectively reduced soil erosion and have successfully restored some degraded ecosystems in this area, where the representative vegetation includes woody plants such as *Robinia pseudoacacia*, shrubs as *Hippophae rhamnoides* and *Caragana microphylla* and herbaceous plants such as *Artemisia sacrorum* and *Stipa bungeana* (*Sun, Liu & Xue, 2016a*). The wood land area has increased from <5% to >40% since 1980 (*Xu, Wang & Zhao, 2018*) (Fig. 1).

## Sampling and data collection

Based on the investigation of the history of land use, we selected four types of vegetation rehabilitation of cropland abandoned between July 10 and September 10 in 2011 and 2012, the season in which plant community biomass peaks. The four types were naturally revegetated grassland (with rehabilitation times of 2yr, 5yr, 8yr, 11yr, 15yr, 18yr, 26yr

and 30yr), planted shrubland (with rehabilitation times of 5yr, 10yr, 20yr, 30yr, 36yr, 47yr), planted woodland (with rehabilitation times of 5yr, 10yr, 20yr, 37yr and 56yr), and orchard land (with rehabilitation times of 5yr, 10yr and 20yr). All these sites suffering similar farming practices before conversion, and the farmlands had more than 200 years of cultivation history (*Deng et al., 2016*). Thus, sloping cropland sites were selected as representative of the original condition for the rehabilitation chronosequences of the naturally revegetated grassland, revegetated shrub land, woodland and orchard land. In this area, the climax vegetation is the *Quercus liaotungensis Koidz* (*Zhang et al., 2011b*), which were naturally regenerated on abandoned land from grassland to shrub land and climax forest (*Q. liaotungensis*) over about 150 years, based on previous research of secondary forests in this area. So, we considered it as representative of the soil-dominated climax community in vegetation rehabilitation to assess the effectiveness of vegetation rehabilitation. These selected sites offered representativeness, typicality and consistency and had similar slope gradients, slope aspects, and topography. The properties of the experimental sites are shown in Table 1.

The size of the plots were varied with the plant communities to match their spatial distribution: the replicated plots of 20 × 20 m were established in each site of planted woodland (*Robinia pseudoacacia*), while the replicated plots of 10 × 10 m were established in each shrub land site (*Caragana microphylla, Hippophae rhamnoides*) and in each orchard land site. The smaller replicate plots (2 × 2 m) were randomly established in each naturally revegetated grassland site (including *Artemisia sacrorum, A. capillaries, A. giraldii, Aneurolepidium dasystachys, Bothriochloa ischaemum, Heteropappus altaicus, Lespedeza bicolor, Stipa bungeana, Setaria viridis*, and other grasses). The plots were separated by at least 50 m.

We choose four random sampling to avoid the sampling error. At each sampling plot, after removing ground litter, five soil layers (0–10, 10–20, 20–30, 30–50, and 50–100 cm) were separately collected with a soil drilling sampler (4 cm diameter). The soil samples from the same layer of the same plot were mixed to form one sample. The samples were divided into two parts and were passed through 2 mm screens for removing roots, gravel, and coarse fragments. Then each sample was brought to laboratory. One part was naturally air-dried to measure the organic carbon and analyse soil organic carbon (SOC), total nitrogen (TN), and total phosphorus (TP) contents, particle size distributions and micro-aggregates. The other part was stored in a refrigerator at 4 °C to analyse water-soluble amounts (carbon, nitrogen), microbial biomass (carbon, nitrogen), enzyme activity as well as other variables not reported in this paper.

## Physical and chemical analyses

The soil bulk density (BD) of each soil layer was measured with the cutting ring method (*Ding et al., 2019*). SOC was determined using the dichromate oxidation method (*Nelson & Sommers, 1982*), and TN was determined using the Kjeldahl method (*Bremner & Mulvaney, 1982*). For soil PSD (particle-size distribution) and micro-aggregate analysis, soil samples were analysed by a laser diffraction technique using a Longbench Mastersizer 2000 (Malvern Instruments, Malvern, England) (*Xiao et al., 2014*). There are some differences between
**Table 1 Basic information of sample plots.**

| Restoration pattern | Site code | Repeat number | Rehabilitation years (a) | Altitude (m) | Slop (°) | Vegetation coverage (%) | Vegetation |
|---|---|---|---|---|---|---|---|
| **Crop land** | AS0 | 3 | 0 | 1,270–1,290 | 17–24 | 32 | *Setaria italica, Glycine max* |
| | AS1 | 3 | 2 | 1,101–1,276 | 13–27 | 12.1-19.8 | |
| | AS2 | 3 | 5 | 1,185–1,262 | 17–19 | 30.7–57.3 | *Geranium wilfordii Maxim, Artemisia leucophylla, Lespedeza bicolor* |
| **Naturally revegetated grassland** | AS3 | 9 | 8 | 1,235–1,276 | 12–40 | 18–60.4 | *Turcz, Tripolium vulgare, Artemisia capillaris, Parthenocissus* |
| | AS4 | 3 | 11 | 1,198–1,292 | 23–37 | 24–76.3 | *tricuspidata, Poa sphondylodes, Leymus secalinus, Stipa bungeana,* |
| | AS5 | 3 | 15 | 1,291–1,306 | 14–19 | 39.8–76 | *Setaria viridis Sonchus oleraceus L, Potentilla bifurca, Cleistogenes* |
| | AS6 | 3 | 18 | 1,179–1,189 | 22–30 | 16–49 | *hancei, Bothriochloa ischaemum, Stipa grandis, Heteropappus altaicus,* |
| | AS7 | 3 | 26 | 1,144–1,161 | 22–28 | 21.8–68.9 | *Dendranthema indicum, Roegneria kamoji* |
| | AS8 | 7 | 30 | 1,149–1,293 | 14–29 | 33–79.7 | |
| | AS9 | 4 | 5 | 1,281–1,290 | 12–21 | 20–38 | |
| | AS10 | 3 | 10 | 1,139–1,161 | 29–32 | 53–78.4 | |
| | AS11 | 3 | 10 | 1,264–1,281 | 14–27 | 36–57 | |
| **Shrub land** | AS12 | 4 | 20 | 1,185 | 21 | 52 | *Hippophae rhamnoides* |
| | AS13 | 3 | 20 | 1,203–1,211 | 21–22 | 28–53 | *Caragana korshinskii Kom.* |
| | AS14 | 3 | 30 | 1,128–1,139 | 14–25 | 21–46.3 | |
| | AS15 | 3 | 36 | 1,211–1,253 | 20 | 46–65 | |
| | AS16 | 3 | 47 | 1,181–1,241 | 18–24 | 49.3–89.6 | |
| | AS17 | 3 | 5 | 1,259–1,288 | 22–34 | 36–56 | |
| | AS18 | 3 | 10 | 1,161–1,227 | 27.5–33 | 38–53 | |
| **Wood land** | AS19 | 3 | 20 | 1,236–1,259 | 17–26 | 32–42 | *Robinia pseudoacacia Linn.* |
| | AS20 | 3 | 37 | 1,209–1,259 | 30–33 | 53–65 | |
| | AS21 | 2 | 56 | 1,170–1,175 | 21–22 | 49–90 | |
| **Orchard land** | AS22 | 3 | 5 | 1,207–1,226 | 0 | | |
| | AS23 | 3 | 10 | 1,220–1,254 | 0 | | *Malus pumila Mill.* |
| | AS24 | 3 | 20 | 1,206–1,222 | 0 | | |
| **Natural forest land** | AS25 | | 100 | 1,332–1,337 | 14–29 | 39–52 | |
| | AS26 | 9 | 100 | 1,235–1,283 | 23–38 | 35–70 | *Quercus wutaishanica Blume.* |
| | AS27 | | 100 | 1,552–1,570 | 28–45 | 10–28 | |

the pretreatment methods for determining soil PSD and micro-aggregate. For soil PSD, soil samples were pretreated with 6% $H_2O_2$ and 10% HCL to remove organic matter and carbonates and oxides and were soaked in distilled water for 24 h, then mechanically dispersed with 0.4% Calgon by an ultrasonic bath for 5 min. For micro-aggregate determination, the soil samples were soaked in distilled water for 24 h and mechanically dispersed in ultrasonication for 5 min (*Xiao et al., 2014*). Soil PSD was described in terms of the percentage of sand (0.05–2 mm), fine silt (0.002–0.020 mm), coarse silt (0.02–0.05 mm) and clay (<0.002 mm). The size grades of the micro-aggregates were classified to be the same as that of the PSD.

### Fractal features

The fractal dimension of the PSD and micro-aggregation were calculated by the following formula (*Tyler & Wheatcraft, 1992*):

$$V(r < R_i)/V_T = (R_i/R_{max})^{3-D}$$

where $r$ is the particle diameter, $R_i$ is the particle size of subinterval $i$ in the particle size grading, $V(r < R_i)$ is the total volume of soil particles with diameter less than $R_i$, $V_T$ is the sum volume of soil particles, and $Rmax$ is the maximum diameter of soil particles.

### Erodibility (K)

Soil erodibility was measured by the K factor in the EPIC model using SOC content and soil PSD (*Williams, Jones & Dyke, 1984*) and was calculated as follows:

$$K = \{0.2 + 0.3\exp[-0.0256\text{SAN}(1-0.01\text{SIL})]\} \times \left(\frac{\text{SIL}}{\text{CLA}+\text{SIL}}\right)^{0.3}$$
$$\times \left(1.0 - \frac{0.25\text{C}}{\text{C}+\exp(3.72-2.95\text{C})}\right) \times \left(1.0 - \frac{0.25\text{C}}{\text{SN1}+\exp(-5.51+22.9\text{SN1})}\right)$$

where SAN, SIL, and CLA are the sand (%), silt (%), and clay (%) fractions, respectively; C is the soil organic carbon content (%); and SNI = 1-SAN/100.

### Statistical analysis

Three-way ANOVA was performed to test the effects of rehabilitation land type (naturally revegetated grassland, woodland, shrub land, orchard land), rehabilitation time (years since sloping cropland abandonment) and soil depth (0–10 cm, 10–20 cm, 20–30 cm, 30–50 cm, and 50–100 cm) on soil mechanical composition and erodibility. Significance was evaluated at the 0.05 level ($P < 0.05$). Duncan's (D) post hoc test was used to perform multiple comparisons when significance of the ANOVA was observed. The differences between the natural forest and various types of vegetation rehabilitation at each last restoration year were examined by student's $t$ test. Pearson correlation analysis was used to analyse the correlations among particle fractal dimension, micro-aggregate fractal dimension, erodibility and soil nutrients, soil texture, and recovery time. In addition, linear regression analysis was used to determine the relationships between each of particle fractal dimension, micro-aggregate fractal dimension, and erodibility and recovery time in each soil layer between 0 and 100 cm. All of the above statistical analyses were conducted using SPSS 17.0 (SPSS Inc., Chicago, IL, USA) and R version 3.4.4.

**Table 2** Tests of the soil particle fractal dimension, micro-aggregate fractal dimension, erodibility linkage with time (years since cropland abandonment), Rehabilitation pattern and soil depth.

| Source | Particles fractal dimension (PD) | | Micro-aggregate fractal dimension (MD) | | Erodibility (K) | |
|---|---|---|---|---|---|---|
| | F | P | F | P | F | P |
| Rehabilitation types | 34.111 | 0.000[***] | 4.901 | 0.002[**] | 10.007 | 0.000[***] |
| Rehabilitation time | 15.283 | 0.000[***] | 0.943 | 0.508 | 5.491 | 0.000[***] |
| Soil depth | 3.282 | 0.012* | 0.108 | 0.98 | 25.056 | 0.000[***] |
| Rehabilitation time × Rehabilitation types | 4.16 | 0.001[***] | 0.628 | 0.678 | 2.127 | 0.062 |
| Soil depth × Rehabilitation types | 0.143 | 1 | 0.568 | 0.867 | 0.393 | 0.966 |
| Soil depth × Rehabilitation time | 0.278 | 1 | 0.321 | 1 | 0.379 | 1 |
| Rehabilitation time × Soil depth × Rehabilitation types | 0.568 | 0.933 | 0.824 | 0.685 | 0.349 | 0.996 |

**Notes.**
[**]Indicates a significant difference at the 0.01 level ($P < 0.01$).
[***]Indicates a significant difference at the 0.001 level ($P < 0.001$).

## RESULTS

Rehabilitation time, and rehabilitation land type had significant effects on the soil PSD fractal dimension and K factor, only the rehabilitation land type had significant effects on the soil micro-aggregate fractal dimension (Table 2). PSD fractal dimension, micro-aggregate fractal dimension and erodibility showed trends of decline since cropland in all land types. However, PSD fractal dimension, micro-aggregate fractal dimension and erodibility varied among the land types (Figs. 2, 3 and 4).

### The rehabilitation pattern in naturally revegetated grassland

Fractal dimensions of PSD and K factor began to show greater decreases in naturally revegetated grassland than in cropland at 5th, 11th years, respectively, since cropland abandonment. And these trends mainly occurred in 0–20 cm, gradually weaken with depth. Overall, the minimum time before significant decreases appeared in the particle fractal dimension, soil micro-aggregate fractal dimension and erodibility varied among the different rehabilitation land types; in general, the times were shorter for naturally revegetated grassland than for the other land types (Figs. 2–4) (Tables 1–3). Naturally revegetated grassland did not differ from natural forest in fractal dimensions of micro-aggregation or K factor (20–100 cm) over rehabilitation time (Figs. 5, 6 and 7) (Tables 1–3). Linear regression revealed that fractal dimensions of PSD decreased with the number of years since farmland conversion in the 0–50 cm (except in 20–30 cm)(Figs. 2–4) (Tables 1–3).

### The rehabilitation pattern in Wood land

Fractal dimensions of PSD, fractal dimensions of micro-aggregation and K factor began to show greater decrease in woodland than in cropland in the 10th, 10th and 37th respectively since cropland abandonment (Figs. 2–4) (Tables 1–3). Over rehabilitation time, woodland did not differ from natural forest in fractal dimensions of PSD (30–100 cm) and fractal dimensions of micro-aggregation (0–100 cm), K factor in woodland did not differ from natural forest (0–30 cm) and even was lower than that in natural forest (30–100 cm)

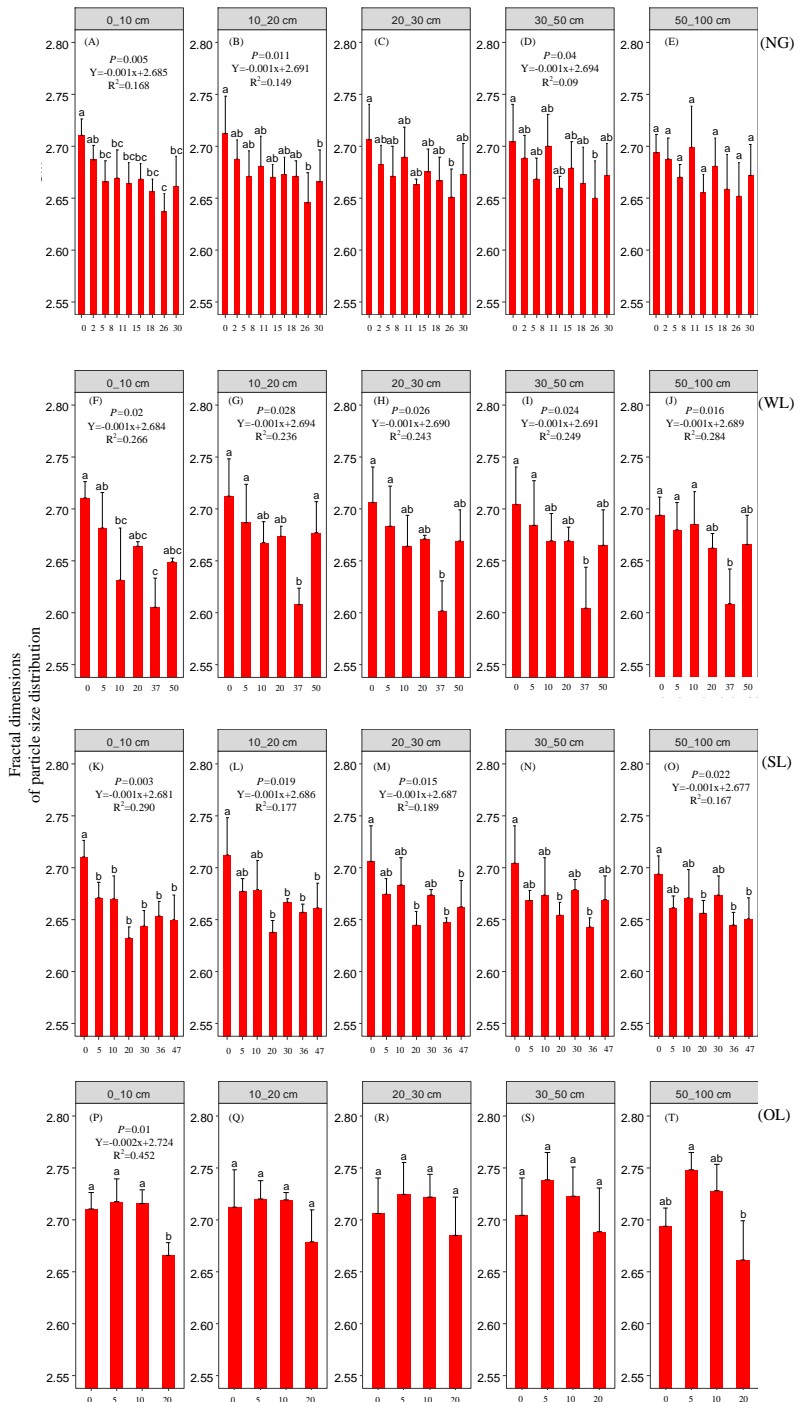

**Figure 2** **Particle fractal dimension change with time since cropland abandonment in various vegetation restoration patterns.** Note: NG, naturally revegetated grassland; WL, woodland; SL, shrub land; OL, orchard land. We set the CL as the initial stage of the rehabilitation process. (A–E) The soil layers of 0–10 cm, 10–20 cm, 20–30 cm, 30–50 cm and 50–100 cm of naturally revegetated grassland, (F–K) the soil layers of 0–10 cm, 10–20 cm, 20–30 cm, 30–50 cm and 50–100 cm of woodland, (L–O) the soil layers of 0–10 cm, 10–20 cm, 20–30 cm, 30–50 cm and 50–100 cm of shrub land, (P–T) the soil layers of 0–10 cm, 10–20 cm, 20–30 cm, 30–50 cm and 50–100 cm of orchard land. Different lower-case letters above the bars mean significant differences among different ages within the same rehabilitation patterns ($P < 0.05$).

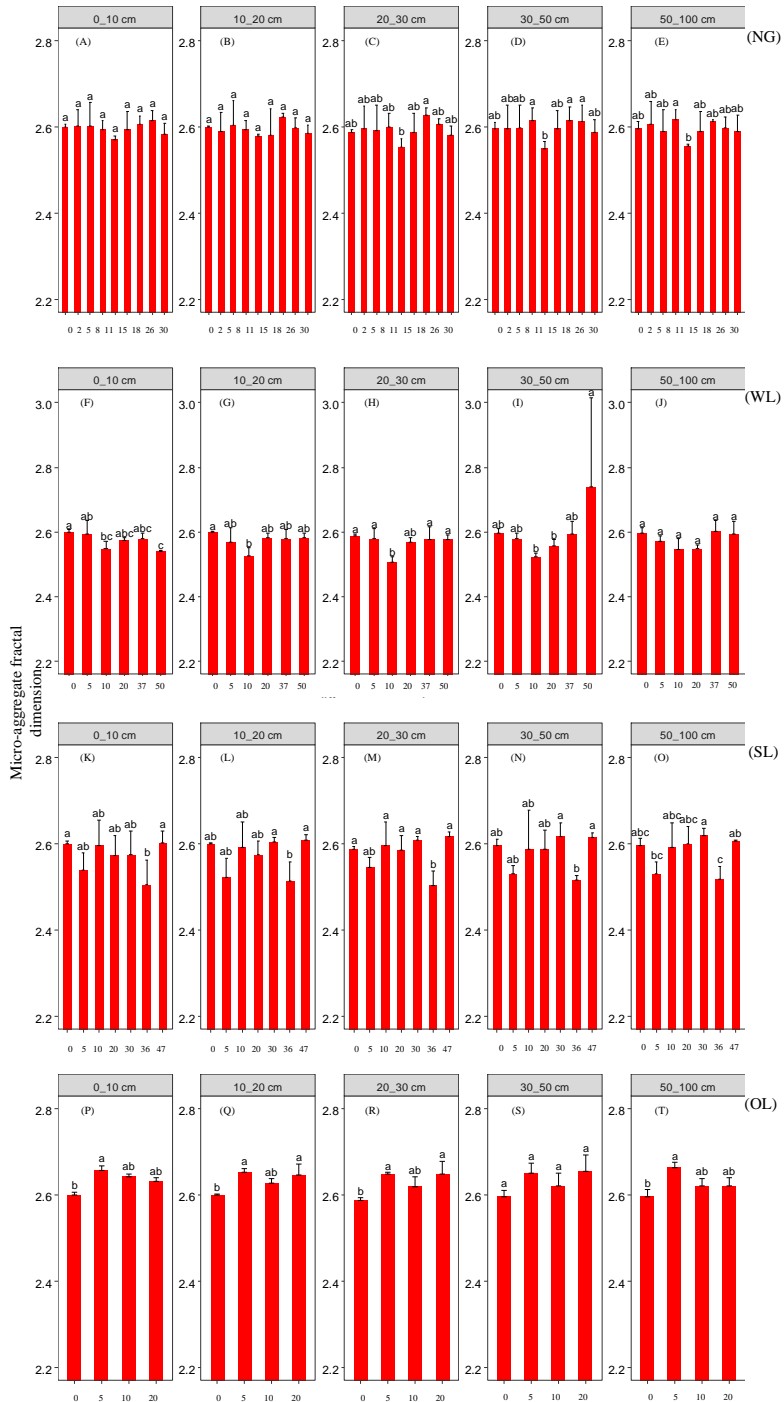

**Figure 3  Micro-aggregate fractal dimension change with time since cropland abandonment in various vegetation rehabilitation patterns.** Note: NG, naturally revegetated grassland; WL, woodland; SL, shrub land; OL, orchard land. We set the CL as the initial stage of the rehabilitation process. (A–E) The soil layers of 0–10 cm, 10–20 cm, 20–30 cm, 30–50 cm and 50–100 cm of naturally revegetated grassland, (F–K) the soil layers of 0–10 cm, 10–20 cm, 20–30 cm, 30–50 cm and 50–100 cm of woodland, (L–O) the soil layers of 0–10 cm, 10–20 cm, 20–30 cm, 30–50 cm and 50–100 cm of shrub land, (P–T) the soil layers of 0–10 cm, 10–20 cm, 20–30 cm, 30–50 cm and 50–100 cm of orchard land. Different lower-case letters above the bars mean significant differences among different ages within the same rehabilitation patterns ($P < 0.05$).

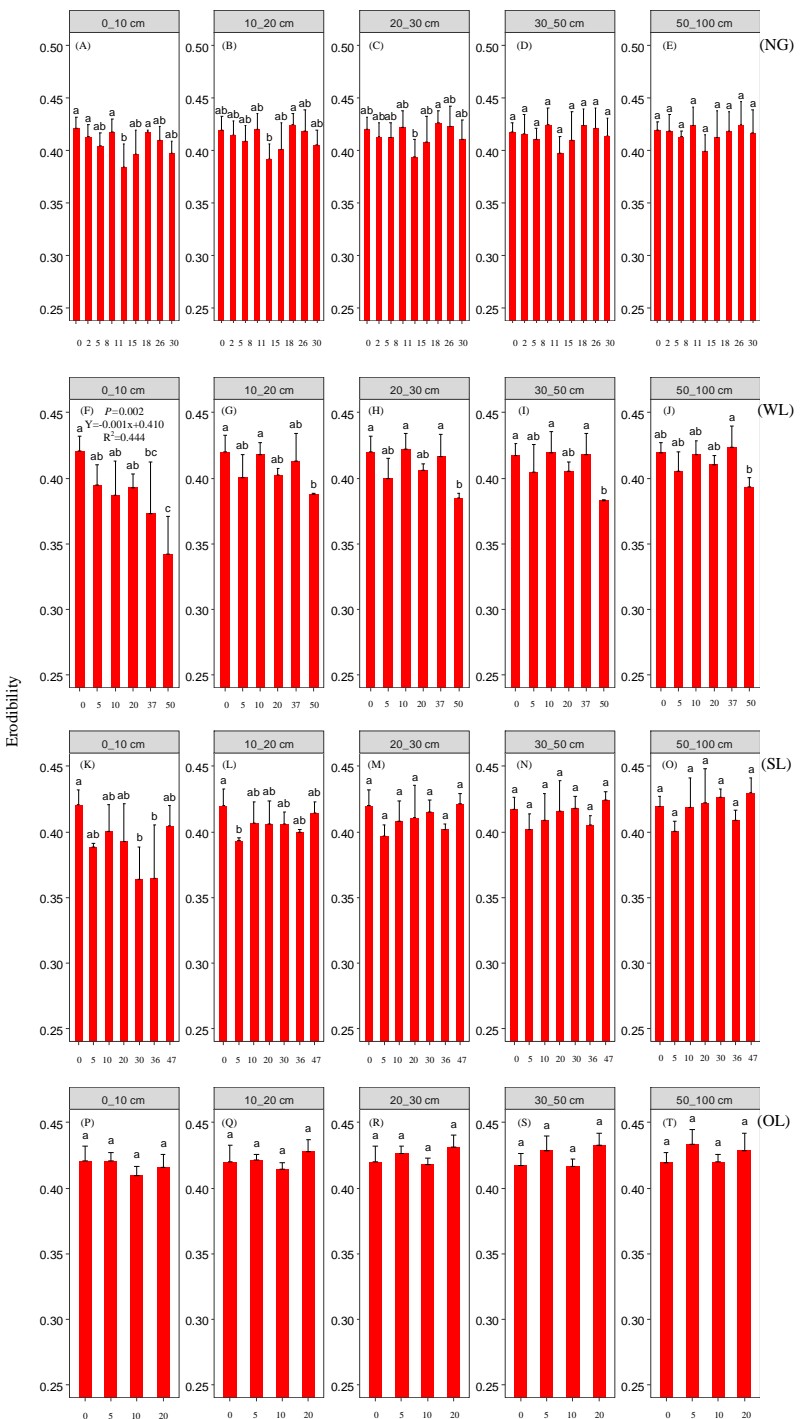

**Figure 4 Erodibility change with time since cropland abandonment in various vegetation rehabilitation patterns.** Note: NG, naturally revegetated grassland; WL, woodland; SL, shrub land; OL, orchard land. We set the CL as the initial stage of the rehabilitation process. (A–E) The soil layers of 0–10 cm, 10–20 cm, 20–30 cm, 30–50 cm and 50–100 cm of naturally revegetated grassland, (F–K) the soil layers of 0–10 cm, 10–20 cm, 20–30 cm, 30–50 cm and 50–100 cm of woodland, (L–O) the soil layers of 0–10 cm, 10–20 cm, 20–30 cm, 30–50 cm and 50–100 cm of shrub land, (P–T) the soil layers of 0–10 cm, 10–20 cm, 20–30 cm, 30–50 cm and 50–100 cm of orchard land. Different lower-case letters above the bars mean significant differences among different ages within the same rehabilitation patterns ($P < 0.05$).

**Table 3** Relationships between particle fractal dimension, micro-aggregate fractal dimension, erodibility and soil nutrients and soil bulk density.

|  | BD | TOC | TN |
|---|---|---|---|
| Particle fractal dimension fractal dimension | 0.185[**] | −0.380[**] | −0.146[**] |
| Micro-aggregate fractal dimension fractal dimension | 0.018 | −0.024 | 0.048 |
| Erodibility | 0.410[**] | −0.658[**] | −0.399[**] |

**Notes.**
[*]Correlation is significant at the $P < 0.05$ level (2-tailed).
[**]Correlation is significant at the $P < 0.01$ level (2-tailed).

(Figs. 5–7) (Tables 1–3). Compared to fractal dimensions of micro-aggregation and K factor, fractal dimensions of PSD in this site showed a clear decreasing tend at the all soil layers (Figs. 2–4) (Tables 1–3).

## The rehabilitation pattern in Shrub land

Fractal dimensions of PSD, fractal dimensions of micro-aggregation and K factor began to show greater decreases in shrub land than in cropland in the 20th, 36th and 30th years, respectively, since cropland abandonment, and tended to have significant differences with cropland since then (Figs. 2–4) (Tables 1–3). The decreasing trend of fractal dimensions of PSD and fractal dimensions of micro-aggregation occurred in 0–100 cm, but it gradually weaken with increasing depth. After rehabilitation, naturally revegetated grassland were also higher than natural forest in fractal dimensions of PSD (0–50 cm), fractal dimensions of micro-aggregation (0–30 cm) and K factor (0–10 cm) (Figs. 5–7) (Tables 1–3), and had no difference with that of natural forest at the deep layer. Linear regression indicated that fractal dimensions of PSD decreased with the number of years since farmland conversion in the 0–100 cm (except 30–50 cm) (Figs. 2–4) (Tables 1–3).

## The rehabilitation pattern in Orchard land

Fractal dimensions of PSD and fractal dimensions of micro-aggregation showed a trend of lower levels in orchard land than in cropland, but there are no significant differences between them (Figs. 2–4) (Tables 1–3). However, K factor in this site didn't decrease after a long-term rehabilitation. Following rehabilitation, fractal dimensions of PSD, fractal dimensions of micro-aggregation, and K factor were significantly higher in orchard land than in natural forest at shallow soil layer (Figs. 5–7) (Tables 1–3). Linear regression revealed that fractal dimensions of PSD decreased with the number of years since farmland conversion in the 0–10 cm soil layers (Figs. 2–4) (Tables 1–3).

## DISCUSSION

### Effects of rehabilitation time on soil mechanical composition and erodibility

In our study, rehabilitation time was a key factor in driving changes in soil mechanical condition, erodibility and properties (Table 2) (Fig. 8). The fractal dimensions of PSD, K factor of the soil in the various rehabilitation land types showed decreasing trends following rehabilitation (Figs. 2–4) (Tables 1–3). These changes were mainly due to the large amounts

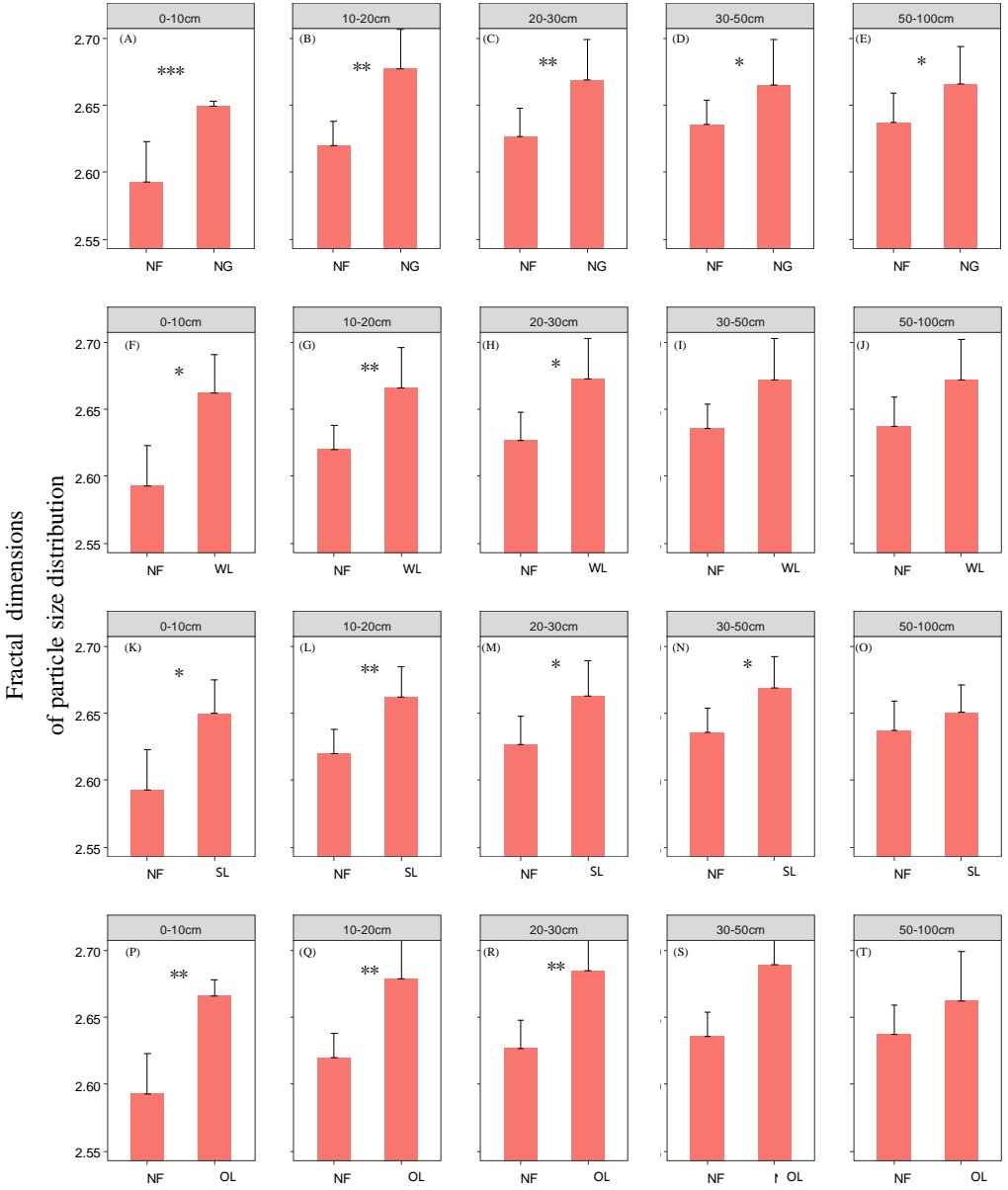

**Figure 5** **The differences of particle fractal dimension between the natural forest and various vegetation restoration patterns at each last restoration year.** Note: NG, naturally revegetated grassland; WL, woodland; SL, shrub land; OL, orchard land; NF, natural forest. (A–E) The soil layers of 0–10 cm, 10–20 cm, 20–30 cm, 30–50 cm and 50–100 cm of naturally revegetated grassland, (F–K) the soil layers of 0–10 cm, 10–20 cm, 20–30 cm, 30–50 cm and 50–100 cm of woodland, (L–O) the soil layers of 0–10 cm, 10–20 cm, 20–30 cm, 30–50 cm and 50–100 cm of shrub land, (P–T) the soil layers of 0–10 cm, 10–20 cm, 20–30 cm, 30–50 cm and 50–100 cm of orchard land. * means significant differences between the natural forest and various vegetation restoration patterns at each last restoration year ($P < 0.05$).

of soil nutrients released by residues and decomposing dead roots, and they promote plant growth and rehabilitation succession (*Guo et al., 2013*). Fractal dimensions of PSD, K factor positively correlated with SOC, and the improvement of mechanical conditions were

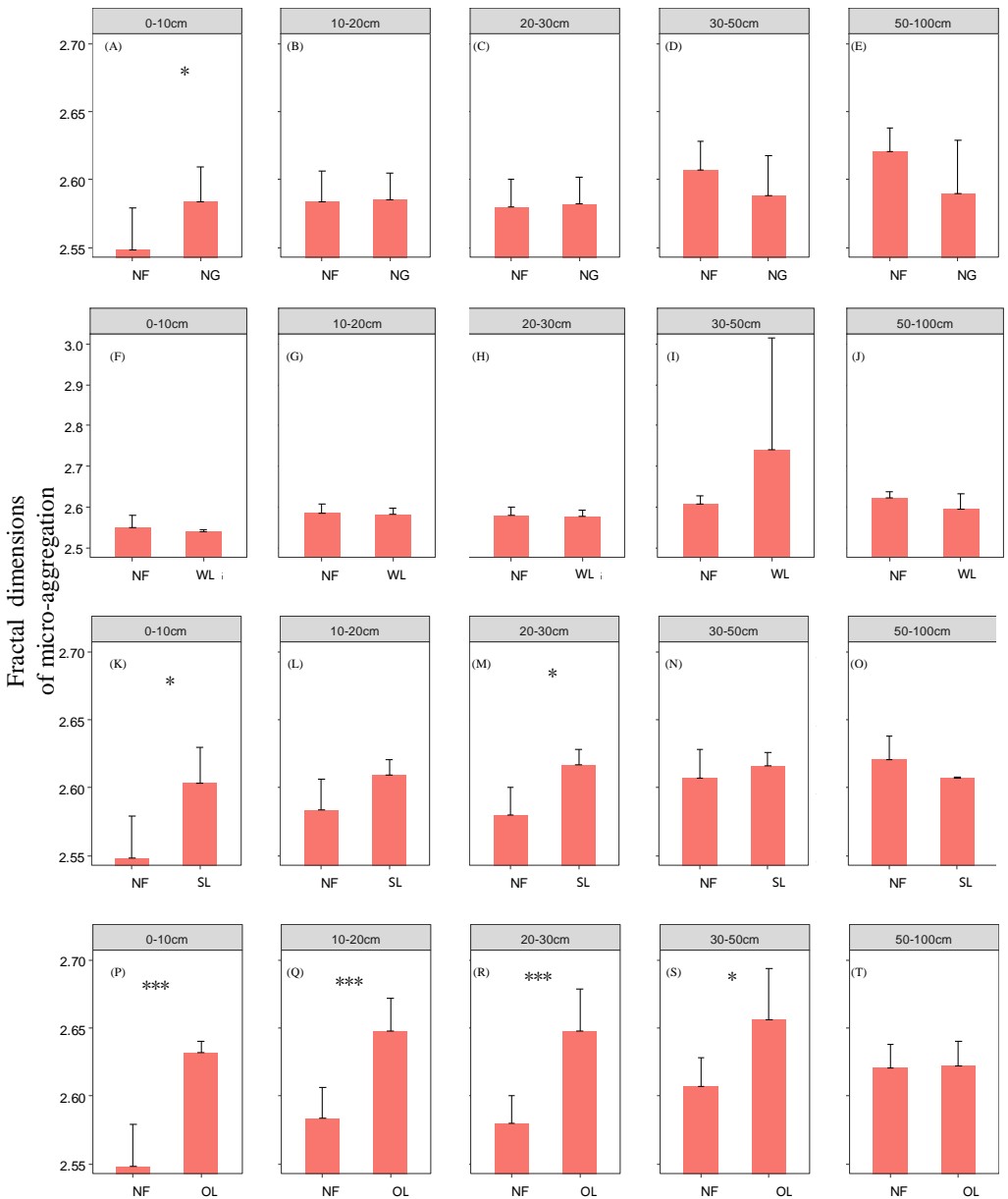

**Figure 6   The differences of micro-aggregate fractal dimension between the natural forest and various vegetation restoration patterns at each last restoration year.** Note: NG, naturally revegetated grassland; WL, woodland; SL, shrub land; OL, orchard land; NF, natural forest. (A–E) The soil layers of 0–10 cm, 10–20 cm, 20–30 cm, 30–50 cm and 50–100 cm of naturally revegetated grassland, (F–K) the soil layers of 0–10 cm, 10–20 cm, 20–30 cm, 30–50 cm and 50–100 cm of woodland, (L–O) the soil layers of 0–10 cm, 10–20 cm, 20–30 cm, 30–50 cm and 50–100 cm of shrub land, (P–T) the soil layers of 0–10 cm, 10–20 cm, 20–30 cm, 30–50 cm and 50–100 cm of orchard land. * means significant differences between the natural forest and various vegetation restoration patterns at each last restoration year ($P < 0.05$).

mainly explained by the soil nutrient levels (Table 3). Soil organic matter, as a binding agent, favoured soil structure stabilization and infiltration and protected it from erosion (*García-Orenes et al., 2012*). In addition, well-developed root systems played a vital role in

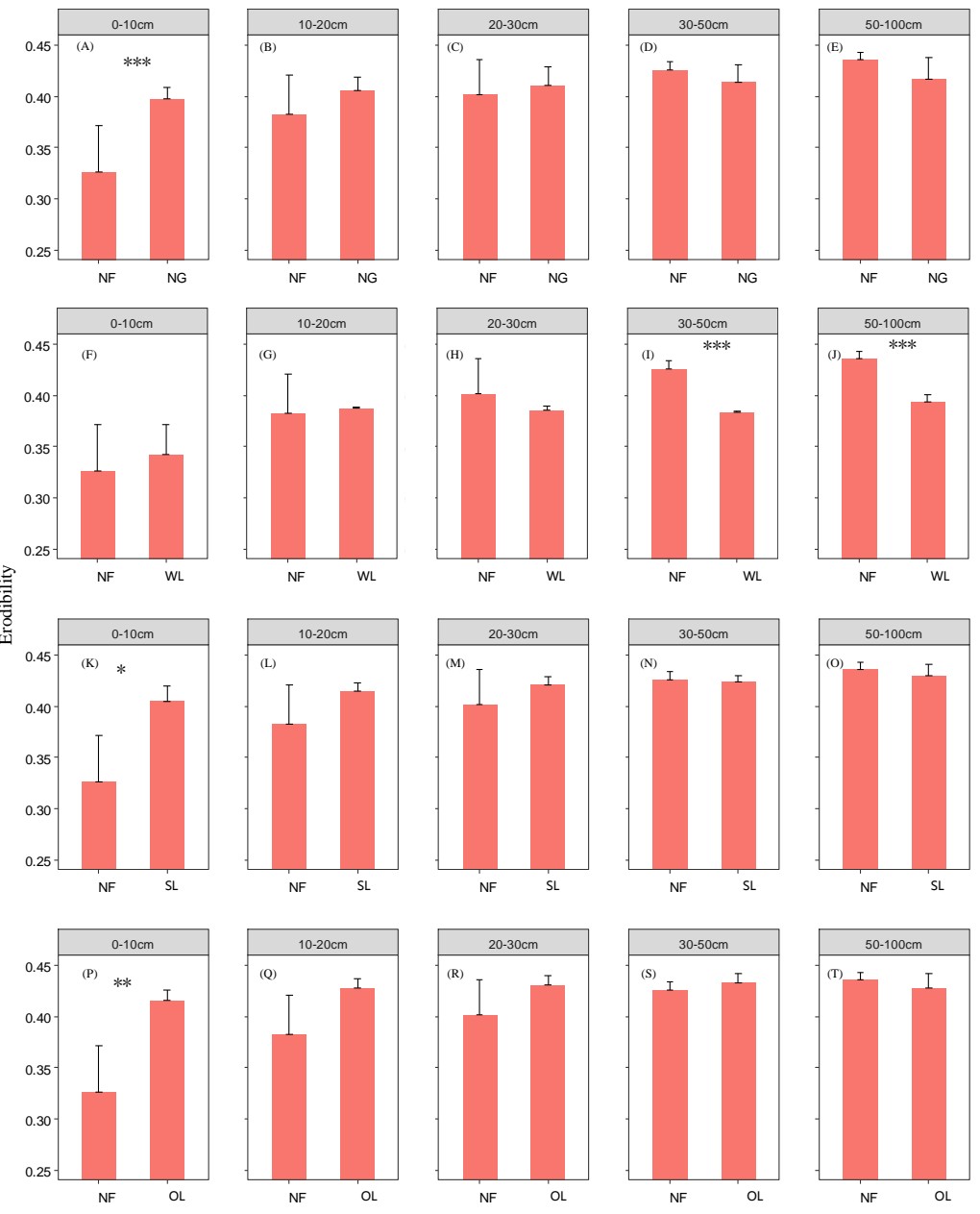

**Figure 7** **The differences of erodibility between the natural forest and various vegetation restoration patterns at each last restoration year.** Note: NG, naturally revegetated grassland; WL, woodland; SL, shrub land; OL, orchard land; NF: natural forest. (A–E) The soil layers of 0–10 cm, 10–20 cm, 20–30 cm, 30–50 cm and 50–100 cm of naturally revegetated grassland, (F–K) the soil layers of 0–10 cm, 10–20 cm, 20–30 cm, 30–50 cm and 50–100 cm of woodland, (L–O) the soil layers of 0–10 cm, 10–20 cm, 20–30 cm, 30–50 cm and 50–100 cm of shrub land, (P-T) the soil layers of 0–10 cm, 10–20 cm, 20–30 cm, 30–50 cm and 50–100 cm of orchard land. * means significant differences between the natural forest and various vegetation restoration patterns at each last restoration year ($P < 0.05$).

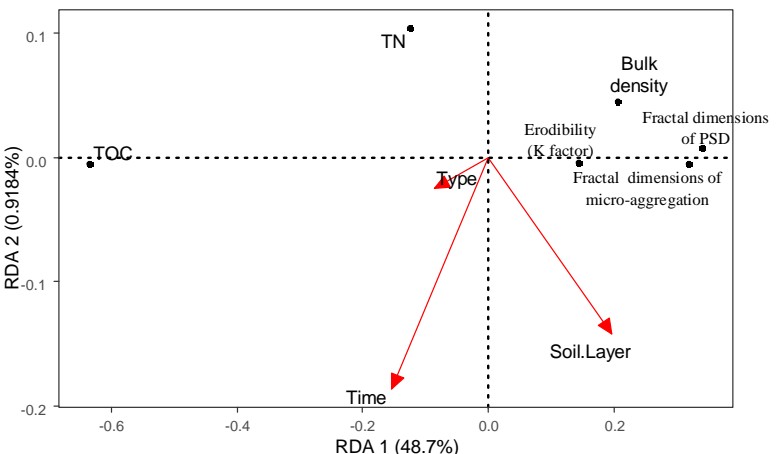

**Figure 8** Biplot of the first two RDA axes between the vegetation Rehabilitation patterns, soil depth, restoration time and fractal dimension, erodibility (K factor), TC, TOC, TN, each classes particles and micro-aggregate.

soil mechanical functioning and actively exude substrates, such as polysaccharides, phenolic compounds, and polygalacturonic acid that affect soil particle cohesion and aggregation (*Hodge et al., 2009*; *Sun, Liu & Xue, 2016b*). Furthermore, by directly binding soil particles *in situ*, plant roots prevented soil from being blown or washed away (*Reubens et al., 2007*). Dense canopies and ground litter following rehabilitation protected soil aggregates from breakdown and prevent particles from being washed away by raindrop energy and runoff (*Zuazo & Pleguezuelo, 2009*; *Wang et al., 2008*).

## Effects of rehabilitation land type on soil mechanical composition and erodibility

Vegetation rehabilitation type was the most influential factor in driving soil mechanical composition and erodibility (Table 2) (Fig. 8). Variation in vegetation recovery patterns and plant traits led to variation in the production and release of soil organic matter, which affect soil crusting, splash, aggregate size and stability. For example, high levels of soil organic matter reduced soil vulnerability to detachment by surface flow, rain splash and other erosion-inducing phenomena (*Xiao et al., 2014*; *Fu et al., 2015*). Litters with different chemical composition among different plant species would impact decomposition rates and the release of soil organic matter (*Ayres, Dromph & Bardgett, 2006*). Thus, naturally revegetated grassland site dominated with high quality litter showed faster circulating rates than shrub and tree sites dominated with relative low quality litter. This phenomenon may explain why the recovery time needed to reach significant improvements of soil mechanical conditions and soil erosion was shorter for naturally revegetated grassland than for woodland and shrub land. Our results are in accordance with the study of *Yu et al. (2015)*, which showed that high concentrations of soil organic matter greatly affected the fractal dimensions of PSD and generally facilitated the improvement of soil structure. The species of naturally revegetated grassland on the whole corresponds with the native

vegetation specieson on Loess Plateau during a long historical period (*Lü, Liu & Guo, 2003*; *Jiang et al., 2013*). They belong to Poaceae and Asteraceae families which are tolerant to drought, cold and grazing due to the characteristics of low water requirements, fibrous root system, are fully suited to the local arid or semi-arid climates and soil (*Lü, Liu & Guo, 2003*; *Jiang et al., 2013*). Thick loess was mainly caused by the loosely cemented silt (*Liu, 1985*; *Yang & Ding, 2008*) which allows rainwater to infiltrate quickly (*Yang et al., 2012*). Thus, naturally revegetated grass as native Loess Plateau vegetation were the best selected species for rehabilitation of soil conditions. *Cespedes-Payret et al. (2012)* even found that afforestation with fast growing exotic species showing its negative effects on soil, compared to native grassland.

In addition, vegetation alleviates erosion of soil by its canopy effectively reducing water-induced soil erosion (*Mohammad & Adam, 2010*; *Wei et al., 2010*). However, that effectiveness was different from various land types. In naturally revegetated grassland site, the lower vegetation layer was more effective in reducing the kinetic energy of rainfall striking the soil surface than the tall vegetation in shrub land and woodland. Owing to the lack of roots at deep soil layer, the naturally revegetated grassland only showed positive effect at the shallow soil layer (0–10 cm). However, due to the stronger stretching ability of the trees roots, woodland site also showed the potential of alleviate soil erosion at deep layer. The soil loss in orchard land was continued over a long time owing to human disturbances (such as production management and tillage practices) and the absence of surface cover protection. This leads to the breakdown of shallow soil aggregates and the washing away of soil particles by raindrop energy and runoff (*Wang et al., 2008*).

**Effects of soil depth on soil fractal dimension and erodibility**

In the analysed four types of vegetation rehabilitation, soil depth had large influences on soil mechanical composition and erodibility (Table 2) (Fig. 8) being consistent with previous studies (*Xiao et al., 2014*). In our study, the positive effects of vegetation recovery mainly occurred in the topsoil with the higher reduction rates of fractal dimensions of PSD, fractal dimensions of micro-aggregation and K factor in the topsoil than in the subsoil (Figs. 2–4) (Tables 1–3). This pattern was resulting from the variation in plant root distribution density decreasing along soil depth (*Reubens et al., 2007*); thus, the deeper soil layers were, the weaker the improvements of soil conditions were (*Sun et al., 2014*). In addition, soil nutrients accumulated near the soil surface due to the decomposition of vegetation litter and by influence of the biogeochemical cycling (*Wang et al., 2014*).

## CONCLUSION

Our study suggested that vegetation rehabilitation time, type and soil depth significantly affect soil mechanical composition and erosion. Following the conversion of sloping cropland to naturally revegetated grassland, shrub land or woodland, the soil structure gradually recovered, and the resistance of the soil against erosive forces gradually increased, primarily within the topsoil. For the conversion of sloping cropland, the natural restoration process of grass represents a more efficient rehabilitation practice than does the planting of other vegetation types. The key point of success rehabilitation project is whether the

selection of species fit current climatic and geological conditions, such as the naturally revegetated grass in our study. Based on the differences of rehabilitation effectiveness among the rehabilitation land type, it is important to carefully select land types for the rehabilitation of soil conditions in the Loess Plateau. Our study, conducted at the regional scale, revealed the effects of vegetation rehabilitation on soil erosion in the Loess Plateau, China, but it strongly contributes to our understanding of the mechanisms through which rehabilitation improves soil quality and provides a suggestion for ecosystem management in arid and semi-arid regions.

## ACKNOWLEDGEMENTS

We thank the anonymous referees and editors of the journal who provided valuable comments and suggestions on our manuscript.

### Funding

This research was funded by the National Key Research and Development Program of China (No. 2016YFC0501707), the Fund Project of Shaanxi Key Laboratory of Land Consolidation (2019-JC15), and the National Natural Science Foundation of China (41771557). The funders had no role in study design, data collection and analysis, decision to publish, or preparation of the manuscript.

### Grant Disclosures

The following grant information was disclosed by the authors:
National Key Research and Development Program of China: 2016YFC0501707.
Fund Project of Shaanxi Key Laboratory of Land Consolidation: 2019-JC15.
National Natural Science Foundation of China: 41771557.

### Competing Interests

The authors declare there are no competing interests.

### Author Contributions

- Leilei Qiao and Wenjing Chen conceived and designed the experiments, performed the experiments, analyzed the data, prepared figures and/or tables, authored or reviewed drafts of the paper, approved the final draft.
- Yang Wu, Hongfei Liu and Jiaoyang Zhang performed the experiments, analyzed the data, prepared figures and/or tables, authored or reviewed drafts of the paper, approved the final draft.
- Guobin Liu and Sha Xue conceived and designed the experiments, contributed reagents/materials/analysis tools, authored or reviewed drafts of the paper, approved the final draft.

### Data Availability

  Raw data is available in the Supplemental Files.

## Supplemental Information

Supplemental information for this article can be found online at http://dx.doi.org/10.7717/peerj.8090#supplemental-information.

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
