# Peer review of "Rehabilitation time has greater influences on soil mechanical composition and erodibility than does rehabilitation land type in the hilly-gully region of the Loess Plateau, China"

_PeerJ, doi:10.7717/peerj.8090_

## Round 0.1 · original submission · Major Revisions

· Academic Editor

Major Revisions

Dear authors,

We have now two review reports that found your manuscript a very interesting contribution to the regeneration of land subjected to the pressure of intense agriculture. They leaved however some constructive suggestions to improve it. The strongest criticism from the reviewers was focused into the use of too many abbreviations along the text and they rather would prefer that you provide the complete terms at least in the discussion, where it turns difficult making a fluid reading of the text. You should also provide more information about the methodology in which you based the study and support your results (see carefully the reports from the two Reviewers that commented and requested changes on this issue). Reviewer 2 has made important observations about the quality and pertinence of your figures. Pay attention to the recommendations and fix the problems.

My main concerns also related to the procedures that you describe for the rehabilitation of abandoned crop lands placed on the Loess Plateau area at the west region of China. While reading the manuscript, I realized that you give not enough attention to describe the original environmental conditions that prevailed at the Loess Plateau region before the development of the intense agriculture time or the type of vegetation that dominated the different ecosystems. That is not detailed in the introduction and it is important to have this reference for a better understanding of your results, particularly the use of natural grassland to obtain a faster and apparently more effective recuperation of the soil fertility. Note that it is a relevant concern from Reviewer 1.

By the way you should include a brief description of the Loess Plateau of China, and particularly tell readers about the original fertility of these lands around 3000 years ago, when the farmers colonized the area and settled on. In recent times, soil loss and desertification were the prevalent conditions. Thus, this explains the necessity of recuperation of the soil fertility and biodiversity which is the main objective of this contribution. Thus, please consider to include some brief historic description of the studied area.

You can find other interesting and useful suggestions in the attached annotated pdf files.

That said, I look forward to see your revised manuscript resubmitted soon.

With my best regards,
Graciela

·

Basic reporting

The subject is exciting and will surely be of great importance for the management and recovery of degraded lands. However, it should be returned to the authors for improvement and clarification so that it can be arbitrated correctly. I am very really looking forward to reading, the future improved version that I hope you will send us as soon as possible.
In the first place, the manuscript shows an abusive use of abbreviations, which makes it extremely difficult to read. Sometimes they forget to clarify the abbreviation, or it is written in two ways...
In the second term, and more importantly, the Introduction and Background needs to be deepened in order to understand the context of the processes. The article aims to be published in a magazine that has a global distribution and is not directed to botanists and soil scientists with specialisation in China, or to a reader to know about the area and its historical and environmental processes.
The literature used is only of Chinese authors, although this is fundamental for the bibliographical references of the studied processes, for the Introduction and necessarily in the Discussion, it should be able to be incorporated to authors also of world reference who are not only from the country.
Figure 1, the map of the area, does not allow to get an idea of the location of the area.
The tables must incorporate information to be explanatory and explain the meaning of the abbreviations. In turn, and linked to the list of plant species (exotic or native, tree, shrub, grass) I am not a botanist, and most readers will not be.

Experimental design

They should improve the clarity and explanation of the methodology in order to be able to read the results and discussion and understand their meaning. I suspended the reading and correction in Results.
I ask the authors: Did they describe the natural horizons of the soil (A, B, C) and their subhorizons? Is that they persist in an area as modified as it appears to be the study area. When they take samples in "five soil layers (0-10, 10-20, 20-30, 30-50, and 50-100 cm)", these correspond to what natural soil horizons? Which corresponds to each sample in each plot? Erosion must have been differential... Do they know? Alternatively, the modifications found in the Results may be influenced by them?
Is fundamental the authors be more explicit before being able to continue the review.
Unfortunately, in the version sent for review, the methods are not described in sufficient detail, nor are the information available is detailed to replicate the sampling.

Validity of the findings

To perform an analysis of the results and conclusions, the authors must make an effort to explain the methodology and why they have opted for it, and only in that case, I can proceed to perform the review and analyse the results correctly.

Additional comments

I suggest returning the manuscript to the authors to improve it and encourage the authors to improve it and to explain the methodology clearly.
I also ask you to incorporate the environmental history of that area, what is the type of soil, I see that it can be a fluvisol, but what kind of fluvisol is it? What natural horizons does it have? What is conserved in the original soil area? Alternatively, the soil disappeared, and they are analysing homogeneous sediment.
You also need to explain what was the original vegetation that gave rise to these soils. What kind of vegetation was used to revegetate (here are the scientific names, but they are too many for a reader to take the trouble to consult in each case what is. If it is exotic or native and if it is a tree, or bush or grass?). With what do the horizons sample in the manuscript correspond with the natural horizons of the soil?
Also, some photographs can complement the explanation.

I have annotated the PDF and upload it as part of my review.

Reviewer 2 ·

Basic reporting

The work seeks to find how to achieve a better soil rehabilitation in a region with strong erosion problems. The extent of the work and the amount of information collected make it very interesting and relevant, both for the journal and for the scientific community. This work highlights the relevance of time since rehabilitation begins and vegetation cover type. It has a very well documented background as well as a complete bibliography throughout the manuscript. I would like to thank the authors for sharing of raw data.
The article needs a revision for its writting. There are small mistakes or repeated statements in the same sentences. As an example, in lines 109-118 the statement “ensuring that all three plots within the site had the same rehabilitation time” was written four times in the paragraph.

2-The authors should unify the use of abbreviations such as MD and PD. For PD usage, I recommend the authors to use “PSD fractal dimension” instead, as "PD" is very similar to “PSD” it makes reading less fluent.

3-From figure 2 to 7, each element contains 20 bar graphs in the same color and bringing little information to the discussion. I recommend the authors to transform al this figures to tables or send it to supplementary materials. As they are shown in this manuscript, are not useful for a better understanding of the article. Another option could be to make an x-y dispersion graph, plotting depth in y-axis for each vegetation type. This could aid to a better understanding of the results.
In case of maintaining bar graphs, Figures 5-7 could be reduced to one bar graph for each depth. It´s not necessary or recommended to graph the same value four times.
It is very difficult to understand the X-axis labels. The number appears to be in a strain

4-Line 89_Authors must also provide soil classification using an international soil description key such as FAO's one

5-In line 124-126 the authors should write down R software citation, as well as check this sentence as text "R version 3.4.4" is repeated.

Experimental design

About plot size selection, the authors used different plot size for each vegetation type. But they never explain why they did use different sizes, or how did they arrive to choose that size for each vegetation type. Also, there no mention about how they sample the Slopping cropland or the reference site.

Line 121-122 the authors are describing a composite sample and make no mention of the number of samples mixed.

It should have been mentioned that samples were brought to the lab before the 2mm mesh sieving, and not after.
Also, In line 122 they said samples were air dried, and then sent to the lab. In the lab, sample were split in two, in which one part was refrigerated. I think the authors wanted to say the sample was split in two, in which one part was refrigerated and another air dried, before sieving.

Validity of the findings

No comment

Additional comments

I think it is an interesting work on a very important issue such as erosion in the hilly-gully region of the loess plateau. The reading of the introduction and discussion was very enjoyable, as well as being very well documented. Unfortunately, the graphics made it difficult for me to understand the article rather than facilitate it. I recommend improving the graphics to achieve a greater visual impact and not use so many abbreviations. Although they allow to provide more information, they can make reading difficult if they are many by making the reader look up the description above.

---

## Round 0.2 · Minor Revisions

· Academic Editor

Minor Revisions

Dear authors,

After you submitted the new version of your manuscript I decided to send it for an additional review round because even though I recognized that it is a highly improved version of the originally submitted manuscript (and I am very glad for that), I found that some of the requested changes had not been taken into account. Considering the structure of the manuscript, I clearly noted that I could read it friendly; casually, the assigned reviewer had the same impression. Figures were also improved and an important number of references were also added to the list although some important ones are yet required (see below).

However, in agreement with the reviewer my main concern is the lack of information that you provide to the readers about the history of the studied region in relation to the original plant community that characterized it. I understand that the Loess Plateau of China has been subdued since a long time to different pressures because of the development of the agriculture, but there are some papers where authors intended to reconstruct the original vegetation using paleontological and paleoecological reconstructions as an analogue (see for instance the following papers, and references therein):

Jiang et al., 2013. Chinese Loess Plateau vegetation since the Last Glacial Maximum and its implications for vegetation restoration. Journal of Applied Ecology 2013, 50, 440–448. doi: 10.1111/1365-2664.12052; and

Lu, H., Liu, D. (Liu Tungsheng) & Guo, Z. 2003. Natural vegetation of geological and historical periods in Loess Plateau. Chinese Science Bulletin, 48 (5): 411-416. https://doi.org/10.1007/BF03183240

Considering that you studied four land types of vegetation that are being used for rehabilitation of the Loess Plateau landscape, and you analyzed which one is more effective to reduce erosion and improve the soil fertility and mechanical composition, it is important that you have into account the original natural coverage for to understand and explain better your results. It seems logic to think that the species selection most similar to the original natural landscape will be the most successful to the rehabilitation. That would explain also the different time needed to get a better result.

Therefore, I strongly agree with the reviewer’s recommendations, to which I am very grateful for the twice carefully detailed revision made of your article.

You have to provide information about the history of the Loess Plateau and the ecological pressures to which this landscape was subdue and discuss about if the four land types of vegetation that you studied are more or less related to the original native vegetation or if they are exotic species, please specify that and provide an explanation about the reasons for which these species were selected (e.g. they are fast-growing species, more resistant to the human pressures in the region, etc.).
In sum, I suggest that you follow step by step the suggestions included in the reviewer report that you will receive along to this letter, and I am sure that you will perform a much more striking article within an increasingly worldwide troublesome research area.

I am also providing an annotated pdf file that notes some typo errors and serious concerns about the reference list, which must be strongly improved following the style of PeerJ.

My hope is that you find all these recommendations useful to enhance your work, and you can submit the revised manuscript very soon.

Best regards,
Graciela Piñeiro

·

Basic reporting

As expressed in my first evaluation, the topic is exciting, especially for its potential for the management and recovery of degraded lands. However, I consider that in the current state it cannot be accepted.
I must emphasize that in this second version the manuscript has substantially improved readability, by eliminating most of the unnecessary abbreviations by the authors and having homogenized the terminology, allowing a fluid and comprehensive reading.
However, although I understand that they have made an effort to attend to the suggestions made by the reviewers, the manuscript still does not have the argumentation to be published in an international journal such as PeerJ.
I believe that both the Introduction and the Background must have a deeper development so that the reader can better understand what the context and evolution of the degradation processes are.
I had argued in the previous evaluation that "The article aims to be published in a journal that has a global distribution and is not directed to botanists and soil scientists with specialisation in China, or to a reader to know about the area and its historical and environmental processes"
Figure 1 has improved substantially. But although they have incorporated other authors, they only appear added to the citations, but they do not contribute to the discussion or argumentation.
One topic in special is very important to me and I had suggested that it be taken into account, and I don't see it contemplated in the new version. In other words, is not yet contemplated the clear analysis of the species of plants and the relationships with the restoration process (exotic or native, trees, shrubs, grasses), I am not botanist, and most readers will not be. But obviously the restoration should be related to the plant cover type. There is no analysis that takes into account the different adaptation of the vegetation because it is native or exotic, and because it is a shrub, tree or grass. Each has a different ecological function and a relationship with the soil, and that should be analyzed or discussed.

Experimental design

I consider that in the new version sent, the methods are described in a little more detail, as is the information available to replicate the sampling. About the multiple questions that I raised in my previous review, I do not find answers in the new version, I believe due to constraints of the methodology used by the authors, where they did not seek to answer them.

Validity of the findings

Now that I can read fluently without the abbreviations, analyzing the previous and current version, we feel that we do not find a great difference between the two, they lack a deep discussion of the results or a correlation with the ecological importance that these results would have.

Additional comments

I suggest returning the manuscript to the authors to make a substantial improvement. Although I understand that they have made corrections, I try to encourage the authors to read the article carefully and look for the idea they are trying to convey with it (given they are wanting to assign it to an international community of readers).
The current version, I see it suitable for a regional journal.
The environmental history of that area has not been incorporated, no questions were answered such as: What is conserved in the original area of ​​the soil? I guess the soil disappeared and they are analyzing homogeneous sediments.
Nor is it clearly explained what was the original vegetation that gave rise to these soils. What type of vegetation was used to revegetate (here are the scientific names, but there are too many for a reader to bother to consult in each case what it is: if it is exotic or native and if it is a tree, shrub or grass?)
The built-in photographs do not adequately illustrate and lack legend, I suggest paying attention to the graphic presentation.

---

## Round 0.3 · Minor Revisions

· Academic Editor

Minor Revisions

Dear authors,

It is great for me to announce that having revised your manuscript, I am really happy with the resulting work. I found just some minor typo errors (be careful to find all them) that you may have to fix and then please return the manuscript for my final acceptance. Thanks very much.

Best regards,
Graciela Piñeiro

---

## Round 0.4 · accepted · Accept

· Academic Editor

Accept

Dear authors,

Thank you very much for the resubmission of your article about the recuperation of soils in the Loess Plateau of Central China. I am happy to tell you that the manuscript is now ready for its publication in PeerJ. I am sure that this paper will contribute a lot with the recuperation of this large and important region of China but also it will be useful as a comparative tool in an eventual rehabilitation of abandoned croplands in other countries. Thus, I am grateful for have had the opportunity to handle it. Congratulations!

With my best regards,
Graciela Piñeiro